# Quantitative Scanning Laue Diffraction Microscopy: Application to the Study of 3D Printed Nickel-Based Superalloys



**Guangni Zhou [1], Jiawei Kou [1], Yao Li [1], Wenxin Zhu [1], Kai Chen [1],\* and Nobumichi Tamura [2],\***

1   Center for Advancing Materials Performance from the Nanoscale (CAMP-Nano), State Key Laboratory for Mechanical Behavior of Materials, Xi'an Jiaotong University, Xi'an 710049, China; zhouguangni@gmail.com (G.Z.); kjw9602@163.com (J.K.); xjtuliyao@gmail.com (Y.L.); zwx185@163.com (W.Z.)
2   Advanced Light Source, Lawrence Berkeley National Laboratory, Berkeley, CA 94720, USA
\*   Correspondence: kchenlbl@gmail.com (K.C.); ntamura@lbl.gov (N.T.); Tel.: +1-510-486-6189 (N.T.)

**Abstract:** Progress in computing speed and algorithm efficiency together with advances in area detector and X-ray optics technologies have transformed the technique of synchrotron radiation-based scanning Laue X-ray microdiffraction. It has now evolved into a near real-time quantitative imaging tool for material structure and deformation at the micrometer and nanometer scales. We will review the achievements of this technique at the Advanced Light Source (Berkeley, CA, USA), and demonstrate its application in the thorough microstructural investigations of laser-assisted 3D printed nickel-based superalloys.

**Keywords:** Laue diffraction; microdiffraction; synchrotron; strain/stress measurements; microstructure imaging; nickel-based superalloys

## 1. Introduction

A Laue pattern, consisting of a regular array of diffraction spots (called Laue reflections), is produced on an area X-ray detector when a polychromatic (white or pink) X-ray beam is directed onto a single crystal. The first X-ray diffraction pattern, recorded in 1912 on a photographic plate by Max von Laue, Walter Friedrich, and Paul Knipping when they illuminated with the newly discovered Röntgen Rays a crystal of $CuSO_4$ [1], was a Laue pattern. Laue diffraction was however quickly superseded by monochromatic X-ray diffraction as the interpretation of the latter proved to be more straightforward in view of the famous theory formulated by William H. Bragg and Lawrence H. Bragg in 1913 [2]:

$$2d_{hkl} \sin \theta_{hkl} = \lambda, \tag{1}$$

which shows that by measuring the angular position ($\theta_{hkl}$) of the reflection and knowing the wavelength $\lambda$ of the incident beam, the interatomic spacing $d_{hkl}$ between atomic planes with indices *(hkl)* can be determined. X-ray diffraction is, therefore, an invaluable tool for crystallography, i.e., the determination of the crystal structure of materials. While the angular positions of the diffraction spots hold information on the shape of the crystal unit cell, their intensities are modulated by the atomic decoration within it. The problem with Laue diffraction is that the wavelength associated with each reflection is unknown a priori. Besides, Laue diffraction is at a disadvantage with respect to monochromatic diffraction because the intensity interpretation of Laue spots is difficult, as each Laue spot can be the superimposition of multiple harmonics, and correction parameters are generally wavelength-dependent. Besides, diffraction signals from a monochromatic beam are far cleaner than

from a polychromatic beam thanks to the intrinsic wavelength filtering of unwanted contributions to the background signal. These difficulties have relegated for some 80 years the use of Laue diffraction almost exclusively to the determination of the orientation of single crystals with known structure, without the possibility of exploiting its main distinct advantage: the speed of data collection. Indeed, with a polychromatic radiation, the Bragg condition is fulfilled simultaneously for many reflections and a single Laue pattern would in principle contain most of the information needed to solve a crystal structure, or at the very least determine the shape of the unit cell. Single crystal diffraction using a monochromatic beam requires the sample to be attached to a rotation stage so that multiple patterns at multiple rotation angles can be collected to obtain a usable set of reflections, an inherently slower process.

Then came the introduction of the synchrotron radiation facility in the 1990s, offering orders of magnitudes higher X-ray flux than what a laboratory source could provide, and the highly collimated beam, making it easier to obtain a small pencil beam in the order of a few tens of microns or less. Laue diffraction experienced a revival in two different ways. First, the availability of small beams led to the possibility of using Laue diffraction in a scanning mode to map the crystal orientation of a polycrystalline sample [3]. The increased angular accuracy of area detectors also led to the possibility of mapping crystal sample deformation and strain [4–6]. Second, Laue diffraction was used for structure solution of macromolecules where high redundancy permits the solving of macromolecular configurations in time-resolved experiments, as the problems of harmonics overlap, and wavelength-dependent correction factors were addressed [7]. Structure solution by the Laue method, as developed with X-rays, benefitted neutron single crystal diffraction to the point of becoming the dominant technique, as single crystal data collection with a monochromatic neutron beam is notoriously slow [8]. Laue neutron single crystal diffraction is, for instance, especially useful in macromolecular and chemical crystallography to directly locate hydrogen atoms that are difficult to see with X-rays [9].

The first development greatly benefitted from the combined progress in highly efficient achromatic X-ray focusing optics and fast area detector technologies. Smaller X-ray beams made possible sample mapping with increasing spatial resolution, while advances in detector technologies provided higher angular resolution and thus higher orientation and strain sensitivity, and higher data collection speed. The X-ray focusing optics of choice for polychromatic hard X-ray (>5 keV) radiation are elliptically-shaped mirrors [10]. Large mirrors that approximate the shape of an ellipsoid such as toroidal mirrors (cylindrical mirrors that are bent sagittally) are often used for prefocusing and create virtual X-ray sources that image the real X-ray source [11]. Final focusing is provided by a pair of orthogonal ultrasmooth mirrors perfectly shaped into ellipses in the Kirkpatrick–Baez configuration (KB mirrors) [12–14]. Nowadays, a 1-μm beam spot has become routine at most synchrotrons, and 100 nm focus has been achieved at several beamlines. It may be important to note that there are many possible focusing optics for monochromatic radiation, such as Fresnel zone plates and compound refractive lenses, but only KB mirrors and X-ray tapered capillaries [15] are truly achromatic and therefore suitable for Laue diffraction.

In this article, we will review the progress made by Laue X-ray microdiffraction as it naturally transitions into a quantitative microstructural scanning imaging tool for material sciences, through its application to the understanding of the microstructure of laser-assisted 3D printed nickel-based superalloys.

## 2. Laser-Assisted 3D Printed Ni-Based Superalloys

Ni-based superalloys are a class of metallic materials combining a range of properties such as high-temperature strength, toughness, and resistance to corrosion and oxidation, which make them widely employed to fabricate structural parts that serve under extreme conditions, such as gas turbine blades, blisks, and vane seal segments for aeronautic engines and nuclear power generating systems [16,17]. The essential solutes in Ni-based superalloys are Al and Ti, with a total concentration of less than 10 at %. The equilibrium microstructure of Ni-based superalloys consists of two phases

called $\gamma$ and $\gamma'$. The solid solution phase $\gamma$ has a face-centered cubic lattice with a random distribution of the different species of atoms. By contrast, the $\gamma'$ phase, with the chemical formula of $Ni_3$ (Al, Ti), has a primitive cubic lattice in which the Al and Ti atoms preferentially sit at the corners of the unit cell while the Ni atoms are at the center of each face. Since both phases have a cubic lattice and very similar lattice parameters, the $\gamma'$ phase precipitates coherently in the $\gamma$ matrix, obeying the cube-on-cube orientation relationship. As the $\gamma'$ precipitates grow larger, misfit dislocations may be generated and the $\gamma/\gamma'$ interface becomes semi-coherent. The elevated strength of Ni-based superalloys comes largely from the $\gamma'$ phase as the precipitate strengtheners, with additions from the solid solution strengtheners of many other alloying elements such as Cr, Fe, Co, V, Mo, W, Ta, Nb and so on in both the $\gamma$ and $\gamma'$ phases. To further enhance the creep resistance, Ni-based superalloys have evolved from an equiaxed grain structure to a directionally solidified (DS) columnar grain structure by eliminating the grain boundaries that are transverse to the major stress axis. In DS Ni-based superalloys, the columnar crystal grains grow along the <001> direction, coinciding with the minimum Young's modulus, so that the thermal stress in the turbine blade developed in the process of engine start-up and shut-down is minimized, and the thermal fatigue resistance is improved [18–20]. Although the performance of DS superalloys is not as good as that of single crystals, they are still widely used due to relatively lower costs. Therefore, in this paper, we will focus on the microstructure study of DZ125L, a commonly used DS Ni-based superalloy for fabricating turbine blades and vanes in China. When analyzing the microdiffraction data, the Laue patterns are indexed using the crystal structure of pure Ni.

During service, superalloys may experience damage due to foreign object impact, fretting, and corrosion pitting. In recent years, laser 3D printing has shown promising potentials for precise repair of DS Ni-based superalloy components [21]. As a surrogate of repairing the turbine blade tips fabricated using DS Ni-based superalloys along the <001> crystallographic directions, the repair surface in our samples is parallel to the (001) crystal plane of the substrate. A typical laser 3D printing process is realized by deposition layer by layer. In this study, each cladding layer is deposited by feeding gas-carried DZ125L superalloy powders into the melt pool generated by laser heating at the surface of the substrate and protected from oxidation by an argon atmosphere. The melt pool position moves as the laser scans on the substrate surface, and the previous melt pool position solidifies rapidly. When depositing a new layer, the adjacent as-deposited layer partly re-melts to form a strong metallurgical bond [22]. Due to the high solidification rate and the cyclic thermal loading accompanying the repeated layer deposition, 3D printing produces finer dendrite and $\gamma/\gamma'$ networks [21], higher microstructural defect concentration [23], less element segregation [24], and different strain/stress distribution [25,26], as compared to conventional casting products. Moreover, it is widely recognized that once the combined amount of Al and Ti exceeds 6 wt %, the alloy becomes extremely susceptible to hot cracking in the heat-affected zone (HAZ).

## 3. Laue X-Ray Microdiffraction Beamlines

The first two dedicated Laue X-ray microdiffraction synchrotron beamlines for strain measurements were concurrently developed towards the end of the 1990s at the Advanced Photon Source (APS) (Argonne, IL, USA) [6] and at the Advanced Light Source (ALS) (Berkeley, CA, USA) [27]. They both achieved a 1-$\mu$m X-ray spot size through KB mirrors and used a charge-coupled device (CCD) X-ray camera to collect diffraction patterns. A third effort was also conducted at the National Synchrotron Light Source (NSLS) (Brookhaven, NY, USA) [4] using a tapered capillary focusing optics and a point detector. All three efforts have one initial goal of measuring electromigration-induced stress build-up during temperature-accelerated testing of aluminum interconnects. The stress gradient was measured at the NSLS, providing the first experimental estimate of the electron wind force [4], while at the ALS, evidence of electromigration-induced plasticity was discovered [28]. As the beamlines continued to be developed both at the APS under the leadership of Gene E. Ice and Bennett C. Larson [29,30] and the ALS under Howard A. Padmore, Alastair A. MacDowell, and Jamshed (Jim) R. Patel [31], thus improving optics and diversifying the applications of the

technique to a whole range of material science problems [32–34], new facilities were constructed around the world. In this context, we can cite the 1B2 beamline at the Pohang Light Source (Pohang, South Korea), the VESPERS (Very sensitive Elemental and Structural Probe Employing Radiation from a Synchrotron) beamline at the Canadian Light Source (Saskatoon, SK, Canada) [35], the BM32 beamline at the European Synchrotron Research facility (ESRF) (Grenoble, France) [36], and the nanodiffraction beamline 21A at the Taiwan Photon Source (TPS) (Hsinchu, Taiwan) [37]. All these beamlines have adopted designs that are similar to either the ALS or the APS ones. Laue X-ray microdiffraction experiments have also been conducted on non-dedicated beamlines at the Swiss Light Source (Villigen, Switzerland) [38] and the Diamond Light Source (Didicot, UK) [39], while newly dedicated beamlines are being currently constructed at the Australian Synchrotron (Melbourne, Australia) and the Shanghai Synchrotron Radiation Facility (Shanghai, China).

The Advanced Light Source Laue X-ray microdiffraction project, with its concept originally tested on the bending magnet beamline 10.3.2 [27], had an end-station built in 1999 on the bending magnet beamline 7.3.3 [31,40], sharing about 50% of its time with first the ALS femtosecond program and then the high-pressure diffraction program, before moving in 2007 to be fully dedicated to the superconducting magnet beamline 12.3.2 thanks to an National Science Foundation (NSF) instrumentation grant [41]. The X-ray beam source is refocused first to the entrance of the experimental hutch by a platinum-coated horizontally deflecting toroidal mirror, and where a pair of slits is used as a virtual secondary source adjustable in size. Final focusing of the X-ray beam to a nominal 1-μm size spot on the sample is provided by a pair of in-house developed KB mirrors [13] with an 8:1 and 16:1 demagnification in the horizontal and vertical directions, respectively. Some experiments require a monochromatic beam instead of the broad bandpass (white or pink) beam used for Laue diffraction, and this is provided by inserting in the path of the X-ray beam a four-bounce (111) silicon monochromator consisting of two identical channel-cuts rotating in opposite direction, permitting switching rapidly from white to monochromatic beam while illuminating the sample at the same spot. The superconducting magnet source has a critical energy around 12 keV, and the available energy range in both white and monochromatic mode is 5–24 keV. The sample is placed on a magnetic mount on top of a stack of stages permitting variable sample configuration and automated scans. Diffraction patterns are collected using a Pilatus 1M detector (DECTRIS, Baden-Dättwil, Switzerland) placed on a slide that can be rotated in the plane of diffraction for the possibility of working in either reflection or transmission mode. An additional VORTEX detector (Hitachi, Tokyo, Japan) is used to collect X-ray fluorescence signal for X-ray microfluorescence (μXRF) mapping and micro X-ray absorption near edge structure (μXANES) measurements.

While the ALS beamline 12.3.2 represents the archetype of the first-generation Laue X-ray microdiffraction facility, the TPS X-ray nanodiffraction (XND) 21A beamline has taken the sample environment technology to the next level foreshadowing where the technique might be going in the future. The TPS beamline is on a tapered undulator source that provides at least one order of magnitude more photon flux ($10^{15}$ ph/s) than current similar beamlines. Like beamline 12.3.2, a horizontally deflecting toroidal mirror refocuses the beam further downstream and a monochromatic beam is provided by a four-bounce silicon monochromator. The beamline uses fixed-shaped KB mirrors fabricated by the J-TEC Corporation (Osaka, Japan) to provide a routine beam focus on the sample of less than 100 nm. This is in part achieved by mounting the entire end-station table on six active vibration control systems. The end-station called focus X-ray for micro-structure analysis (FORMOSA) consists of a Pilatus3 6M detector (DECTRIS) (the six-fold increase in pixel number compared to the 1M detector provides more accurate diffraction peak position determination, and thus higher strain and orientation sensitivity) for Laue X-ray diffraction and a silicon drift detector for X-ray fluorescence measurements. A number of in situ tools have been added to the end-station for complementary measurements, including a scanning electron microscope for fine sample imaging, an X-ray excited optical luminescence (XEOL) microscope, and a scanning probe microscope (SPM), as well as sample heating (up to 1300 K) and cooling (down to 100 K) capabilities. The sample hexapod stage providing

both rotation and scanning is equipped with multiple nanoprobes: an atomic force microscope (AFM) tip, a scanning tunneling microscope (STM) tip, and a nanoindenter, providing opportunities for various in situ experiments to measure microstructure under load. The sample is in an ultra-high vacuum environment ($10^{-8}$ Torr).

As will be seen in the next sections, Laue patterns provide a wealth of microstructural information on the sample and with today's combined fast area detector and high photon flux, it can be collected at high speed at a rate of 20 Hz or more. The ultimate data collection speed rate is influenced by multiple factors such as how diffracting the sample is, detector and data storage rate limit, the available photon flux, scanning sample stage speed, and minimum exposure before sample damage occurs. One pattern of a bulk sample material was collected in 100 ms on ALS beamline 12.3.2 and in 10 ms at the TPS beamline 21A, indicating that large area scans of a sample can be performed in a reasonable amount of time, resulting in the collection of thousands of patterns that need to be analyzed. At each Laue microdiffraction beamline it was quickly realized that none of the existing Laue diffraction indexing packages were up to the task, as they were mainly written to determine crystal orientation from a single pattern. This resulted in the separate development of in-house software to analyze the data produced at beamlines, such as the X-Ray Microdiffraction Analysis Software (XMAS) at ALS [42], LaueGo at APS [43], and LaueTools at ESRF [44]. The analysis of thousands of patterns on a regular desktop or laptop can be tremendously time-consuming for low symmetry phases and heterogeneous samples and would be better performed on a cluster. The need for real-time analysis (where data are analyzed as soon as they get collected) drove the development of cluster versions of the XMAS code such as the fast online X-ray micro-diffraction analysis services (FOXMAS) [45] and the XMAS version implemented at the National Energy Research Scientific Computing (NERSC) center at the Lawrence Berkeley National Laboratory.

## 4. The Intensity of a Laue Pattern

When a white or pink X-ray beam hits a crystal, the resulting interference pattern consisting of regularly spaced reflections called a Laue pattern can be collected on an area detector (Figure 1). Such patterns can be tremendously complicated when either high deformation is involved or multiple crystals diffract at the same time, and analysis of such patterns is not always straightforward.

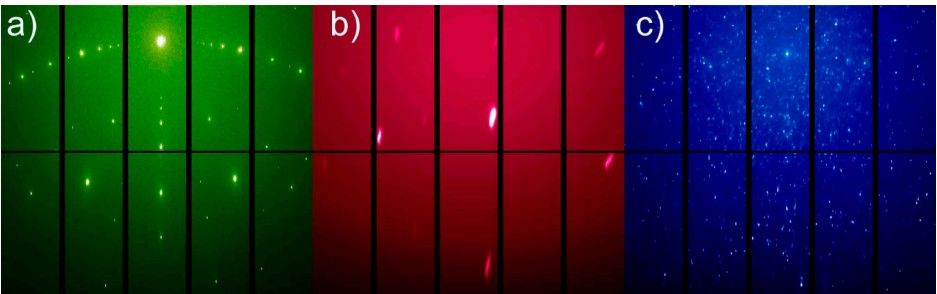

**Figure 1.** Examples of Laue patterns in increasing order of complexity to analyze: (**a**) silicon single crystal; (**b**) a deformed grain of nickel-based superalloy; (**c**) gold film on silicon.

Before going further, however, it is interesting to note that due to the polychromatic nature of the X-ray beam, a Laue pattern captures a way more than diffraction spots. X-ray fluorescence signal as well as inelastic scattering, diffuse scattering, and scattering from the sample environment are also captured and superimposed to the overall background. Effects such as Kossel lines (the interaction of the isotropic X-ray fluorescence signal with the crystal lattice) can take place [46]. All intensities are also modulated by material quantity and absorption. This means that simply recording the overall intensity of a Laue pattern of a Laue X-ray microdiffraction scan can generate maps with interesting topographic and crystallographic values. Consider Figure 2, which shows a map of the average intensity of the

Laue pattern on a section of a 3D printed Ni-based superalloy. In all the following examples shown in this article, the beam size of the X-ray is nominally $1 \times 1\ \mu m^2$.

Dark contrast arises from low intensity, helping highlight areas with no material (such as a crack or a void) or highly defected areas (such as grain boundaries). Differences between peak intensities generate contrast as well. For instance, differences in structure and grain orientation lead to a different number of reflections or a different distribution of reflections, some closer to the center of the detector, others far from the center, leading to crystal grain contrast. In the example shown in Figure 2, crack, secondary phases (precipitates), grains, and subgranular dendrites as well as strain field around precipitates are clearly visible. In this image, intensity filtering using a methodology that has been described in detail elsewhere [47] leads to the enhanced contrast of particular microstructural features of the samples. In Figure 3, the environment around a crack is visualized using the background intensity map (high-intensity pixel belonging to reflections are removed, Figure 3a) and filtered intensity map (Figure 3b) and can be directly compared to a scanning electron microscope image (Figure 3c) taken at the same location.

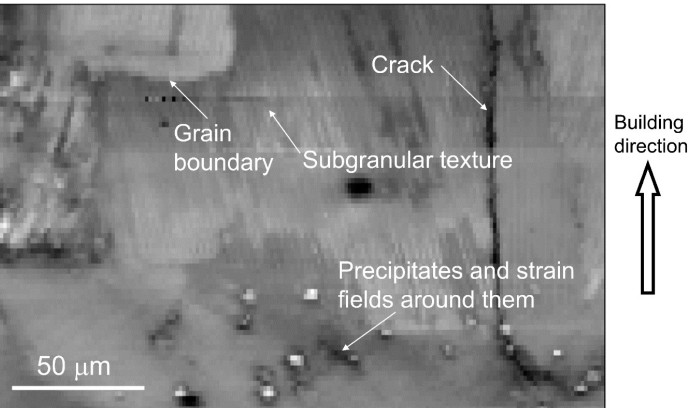

**Figure 2.** A section of a 3D printed nickel-based superalloy as revealed by plotting the filtered intensity of the overall Laue patterns. The bottom part corresponds to the substrate, and the middle and top part to the first few cladding layers. The scanning step sizes along horizontal and vertical directions are 2 and 3 $\mu$m, respectively.

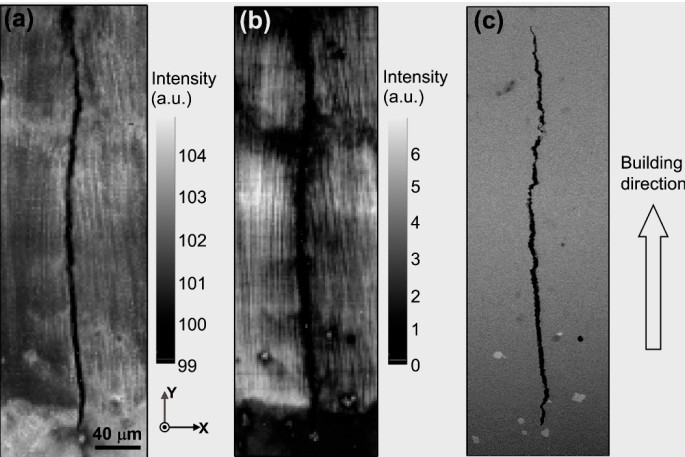

**Figure 3.** A crack originating from the heat-affected zone (HAZ) and propagating in the cladding layer appears clearly in the background intensity image (**a**) and the filtered intensity image (**b**) generated from the Laue diffraction scan. The Scanning Electron Microscope (SEM) image (**c**) does not show such detailed contrast. The scanning step size of the maps is 2 $\mu$m. (adapted from [47]).

## 5. Crystal Structure and Orientation

Back-reflection Laue diffraction [48] has been thoroughly used with laboratory sources to determine the lattice orientation of a single crystal of known structure before its use in other orientation-dependent measurements. The analysis consists in comparing the experimental spot positions collected on the area detector or film with those simulated for given orientations of the crystal. With a submicron size beam, it is now possible to raster scan a polycrystalline sample, collect a Laue pattern at each step, and obtain an orientation map of the sample. In this, the technique of Laue X-ray microdiffraction is similar to electron-backscatter diffraction (EBSD) as it provides high spatial resolution orientation images of the sample and shows the ability to determine the distribution of grain boundary types. However, it differs in the fact that EBSD is a surface technique not probing more than 30 nm below the surface, while hard X-rays can penetrate tens or hundreds of microns deep into the sample. X-ray penetration offers distinctive advantages over electron probes, as buried samples such as interconnects beneath a passivation layer or materials under pressure inside a diamond anvil cell can be probed [49]. It also allows for a variety of sample environments while reducing sample preparation to a minimum: samples can be measured in air, gas, under a vacuum, inside a liquid, and can be heated or cooled, put under high pressure, etc. X-ray Laue microdiffraction can also tackle heavily deformed samples that are challenging to EBSD analysis. Nevertheless, the two techniques have been often used in a complementary manner rather than competitively [50,51], to distinguish surface effects from bulk for instance.

For Laue X-ray microdiffraction scans, the process of indexing the Laue patterns to determine crystal orientations is automated. Codes such as XMAS match experimental angles between reflection pairs with theoretically calculated ones until a minimum number of reflections can be confidently indexed with hkl indices. Criteria such as average angular deviation between the observed and calculated reflection positions and the ratio of the number of reflections found by the number of reflection expected are used to assess the correctness of the indexation. The crystal orientation is then refined using a non-linear least square refinement method. This has been applied to laser-assisted 3D printed nickel-based superalloys to great effect (Figure 4) [26]. The maps show the crystal orientation of nickel at the interface between the last 3D printed epitaxial layers and the first layers of stray non-epitaxial grains. The uniform color in the epitaxial layers indicates that single crystallinity has been essentially preserved in the 3D repair process up to the last layer where stray grains with random orientation occurs.

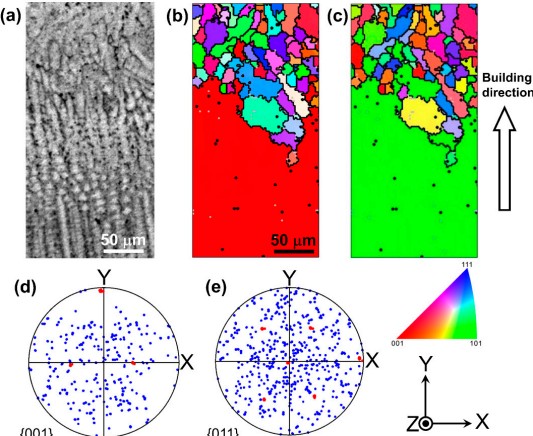

**Figure 4.** Interface between the last 3D printed epitaxial layers and the first stray grain layer. (**a**) Optical microscopy image; (**b**,**c**) inverse pole figure images along the building direction and the normal to the sample surface respectively; (**d**,**e**) corresponding pole figures indicating the orientation of the epitaxial layer (red) and the stray grains (blue) along respectively the 001 and 011 directions. The scanning step size is 2 μm (adapted from [26]).

By comparing crystal orientation matrices measured at adjacent positions (Figure 5), misorientation images of the sample are obtained. This is equivalent to the kernel average misorientation (KAM) maps in EBSD. Such images reveal microstructural substructures such as sub-grain boundaries. In the case of 3D printed nickel-based superalloys, the dendritic structures resulting from the rapid solidification of the molten pool generated by the laser are apparent in the pixel-to-pixel misorientation image (Figure 5a), while the overall trend of increasing misorientation between dendrites originating from different grains with increasing distance from the HAZ can be evaluated by subtracting pixel orientation with fixed orientations in the substrate grains (Figure 5b).

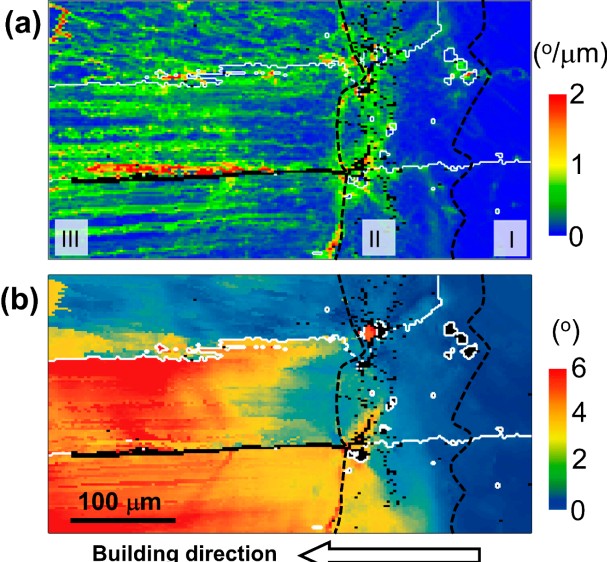

**Figure 5.** Misorientation image of a laser-assisted 3D printed nickel-based superalloy at the interface between the substrate (Region I) and its HAZ (Region II) and the cladding layer (Region III). (**a**) Pixel-to-pixel misorientation; (**b**) misorientation with respect to a fixed orientation in the substrate. The scanning step sizes along horizontal and vertical directions are 3 and 2 μm, respectively (adapted from [52]).

Figure 5 also shows a crack between the bottom and the middle grain that originated in the heat-affected zone and propagated along the grain boundary in the cladding layer. Crack formation and propagation point to the existence of strains. Strains can be readily measured by Laue X-ray microdiffraction, as shown in the next section. Misorientations tend to increase with each passing layer, indicating that there is an accumulation of defects, which is ultimately not good news for the laser-assisted 3D printing process as above a certain threshold of defect concentrations, stray grains will form (Figure 4), destroying the desired single or DS crystallinity. Section 7 will show how defect distribution and concentration can also be mapped by Laue X-ray microdiffraction.

Indexing, especially low symmetry crystals, and multiphase materials, is the most time-consuming process in Laue diffraction analysis and is the reason why several Laue microdiffraction beamlines are investing in clusters, GPU machines, and cloud computing. Although some facilities such as the ALS and TPS offers beamline users access to clusters for real-time analysis of data, many still rely on desktop and laptop machines for data processing. While computing speed has improved greatly over the years and will likely continue to do so in the future, beamline scientists are also increasingly investing in developing better algorithms for data reduction. One way to reduce the number of indexing performed is to first identify grain boundaries in a microdiffraction scan. This can be performed by comparing adjacent patterns for changes in the peak distribution composition. One such algorithm has been recently developed [53]. The analysis is then greatly accelerated by the lesser requirement to index

only one pattern per grain. The remaining patterns need not be indexed as each of the reflections have already been allocated hkl indices.

## 6. Elastic Strains

Both EBSD [54,55] and Laue X-ray microdiffraction [5,42] have been used to measure elastic strains. Both rely on the ability to measure small deviations in the positions of the features resulting from the interaction of the incident beam with the crystal lattice (indirect interaction as Kikuchi lines for the former, direct interaction as Laue reflections or Laue peaks for the latter), with respect to their unstrained positions. These deviations are extremely small and geometry calibration of the experiment must be performed with great care to minimize errors. However, the strain resolution is often limited by geometrical imperfections of the area detector and the algorithm employed in the peak fitting routine. Reflection positions in Laue diffraction, for instance, can only be obtained within a certain level of accuracy limited by the spatial distortion of the detector inherent from its manufacturing process, and by how precisely the shape of the reflection is known. Elastic strains can currently be estimated to within $10^{-4}$ [56] but this can be pushed down to around $5 \times 10^{-5}$ by careful error analysis [57]. The digital image correlation (DIC) technique can also be used to estimate the distortion of the Laue pattern and improve strain resolution to about $10^{-5}$ [58]. Calibrated area detectors with many pixels will likely move the uncertainty further down, once effects such as energy dependence of X-ray penetration have been taken into account.

When it comes to elastic strains, Laue X-ray microdiffraction itself is actually blind to the dilatational component of the strain tensor as only the positions but not the energy of each reflection are accessible a priori [5]. The reflection energy must be measured independently and with good resolution (at least in the order of 1 eV at 10 keV for strain of about $10^{-4}$) by, for instance, scanning the monochromator, using a rainbow filter [59], or using an energy-sensitive detector to obtain the complete strain tensor. However, many experiments studying mechanical properties in the plastic regime are mainly interested in the directions of the elastic deformations and forces (the ways in which the dislocations move), which are readily available in the determination of the deviatoric strain tensor, the quantity measured by Laue X-ray diffraction.

Figure 6 shows deviatoric strain components at the transition between the single crystalline dendritic region and the stray grain regions [26]. It shows that the epitaxial dendritic region is under a tensile strain perpendicularly to the building direction. A tensile strain effectively explains the propagation of cracks in the cladding layers (Figure 5). The anisotropic strain distribution is believed to originate from the layer-by-layer raster scanning deposition process of 3D printing. As fabricated, each freshly deposited layer can be regarded as a free surface, so that the out-of-plane stress, similarly to the case of thin films, is close to zero. Since the temperature of the fresh cladding layer is much higher than of the substrate, in-plane tensile stress is built up in the deposited materials as the material cools down. The strain along the primary dendrite arms is determined by the in-plane stress and has opposite sign to the strain transversely to the dendrite arms, and thus compressive strain is detected along the building direction. A more detailed analysis by our group shows that the defect density close to the columnar-to-stray transition (CET) interface is significantly higher than in the substrate or in the region far from the CET interface. This indicates that dynamic recrystallization is the likely cause of CET [23,26]. The von Mises strain map shows that the plastic strain in the interdendritic regions is higher than in the dendrite cores. Therefore, during the post-processing heat treatment and service at elevated temperatures, rafting or even recrystallization may preferentially occur in the interdendritic regions.

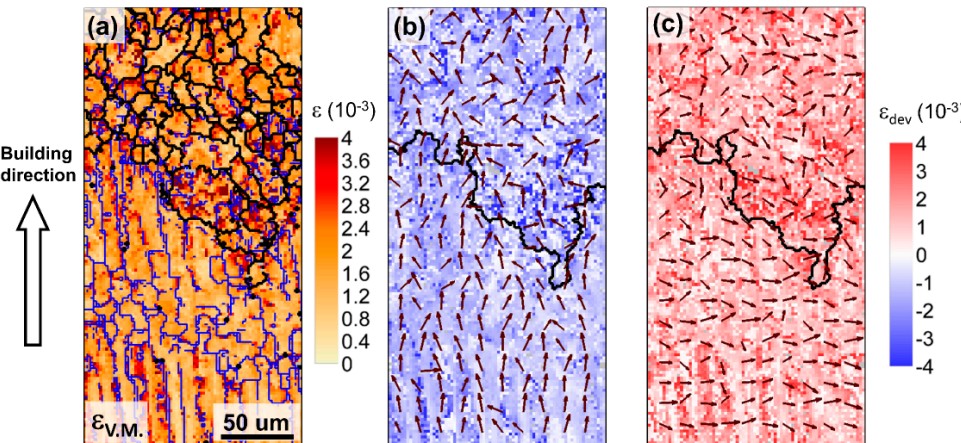

**Figure 6.** (**a**) Equivalent (von Mises) strain ($\varepsilon_{V.M}$); (**b**) value and in-plane projection of the principal axis compressive component of the deviatoric strain ($\varepsilon_{dev}$) tensor; (**c**) value and in-plane projection of the principal axis tensile component of the deviatoric strain tensor. The scanning step size is 2 μm. (adapted from [26]).

## 7. Plastic Strains

While elastic strains (at least the deviatoric part of them) will cause shifts in the Laue peak positions, the accumulation of defects such as dislocations manifests as peak broadening. The shape of the Laue reflections, therefore, contains information on the plastic strain experienced by the material. Several types of dislocation arrangements can be measured by Laue diffraction [60,61]. Geometrically necessary dislocations (GND) constitute arrays of dislocations of the same sign accommodating local bending of the crystal lattice. The latter manifests itself by the anisotropic broadening (streaking) of reflections along directions perpendicular to the dislocation lines. By measuring the angular length of the streaks, GND density can in principle be measured using the Cahn–Nye formula [62,63]:

$$\rho_{GND} = \frac{1}{Rb},$$

(2)

where $R$ is the radius of curvature of the bending and $b$ is the Burgers vector. A similar approach can, in fact, be used in EBSD to measure local GND density [64].

Moreover, one given dislocation slip system often results in peak streaking in specific directions which often allows for precisely determining which slip system participates in the plastic deformation. This is, for instance, the case for face-centered cubic (FCC) systems where the 12 possible slip systems result in different streaking patterns [65].

Statistically stored dislocations (SSD) in contrast result in an angularly isotropic broadening of the reflections. In the presence of GNDs, SSD density can be assessed by measuring the angular width of the reflection perpendicularly to the GND streaking direction, which is a measure of the total dislocation density (GND + SSD).

When dislocations rearranged themselves in an annealing process to form dislocation walls and sub-grain boundaries, the resulting Laue peaks split with each sub-peak corresponding to one sub-grain orientation. These types of boundaries are often referred to as geometrically necessary boundaries (GNBs) as they are made of stacking of GNDs. In a similar way to the analysis of peak streaking, the measure of the angle between two sub-peaks can be used to estimate GND density and the streaking direction reflect what slip system these GNDs belong to. Figure 7 shows the distribution of the average peak width around the HAZ of a laser-assisted 3D printed nickel-based superalloy (same region as Figure 5) [52]. The reflections start by being relatively round and sharp in the substrate, indicating the presence of SSDs only, but show anisotropic broadening in the heat-affected zone,

indicating the formation of GNDs. In the dendritic region, peak splitting is observed, indicating the formation of GNBs.

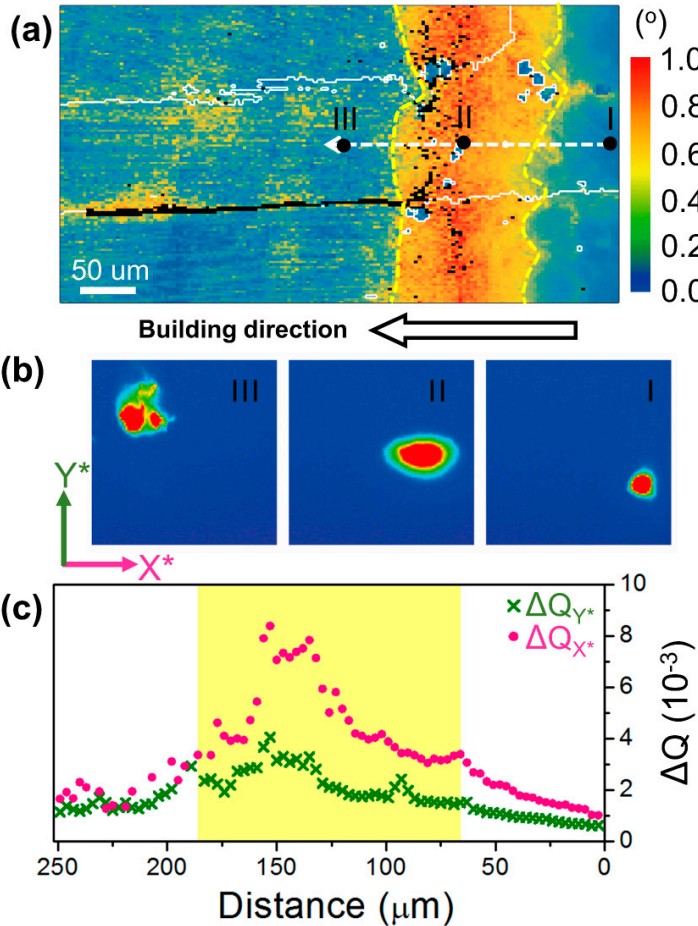

**Figure 7.** (**a**) Average peak width image of the same area depicted in Figure 5; (**b**) Shape of a Laue reflection at position I (substrate), II (HAZ), and III (cladding layer); (**c**) Angular width in reciprocal space of the reflection vs., position along lines I–III of (**a**), in the $X^*$ and $Y^*$ directions. The scanning step sizes along horizontal and vertical directions are 3 and 2 μm, respectively (adapted from [52]).

In Figure 8 [66], in an effort to understand the origin of ductility dip cracking in the stray grain polycrystalline region of a 3D printed nickel-based superalloy, the orientation of each grain on either side of the crack has been mapped (Figure 8a) and the active dislocation slip system has been studied. The approach consists in comparing the experimental streaked Laue pattern (Figure 8b) with those simulated using all the known slip systems (Figure 8c). In this particular case, a match is found between the experimental data and the (111) [1–10] slip system. In all these 40 numbered grains, edge-type GNDs dominate, and their dislocation line directions are almost parallel to the crack plane. Based on Schmid's law, the equivalent uniaxial tensile force direction is revealed normal to the trace of the crack. Thus, it is proposed that because of a significant temperature gradient, thermal tensile stress perpendicular to the laser scanning direction is elevated and exceeds the yield stress. As the dislocations slip inside the crystal grains and pile up at the grain boundaries, local strain/stress keeps increasing, until the materials in these regions fail to sustain further deformation, leading to the formation of voids and propagation of cracks.

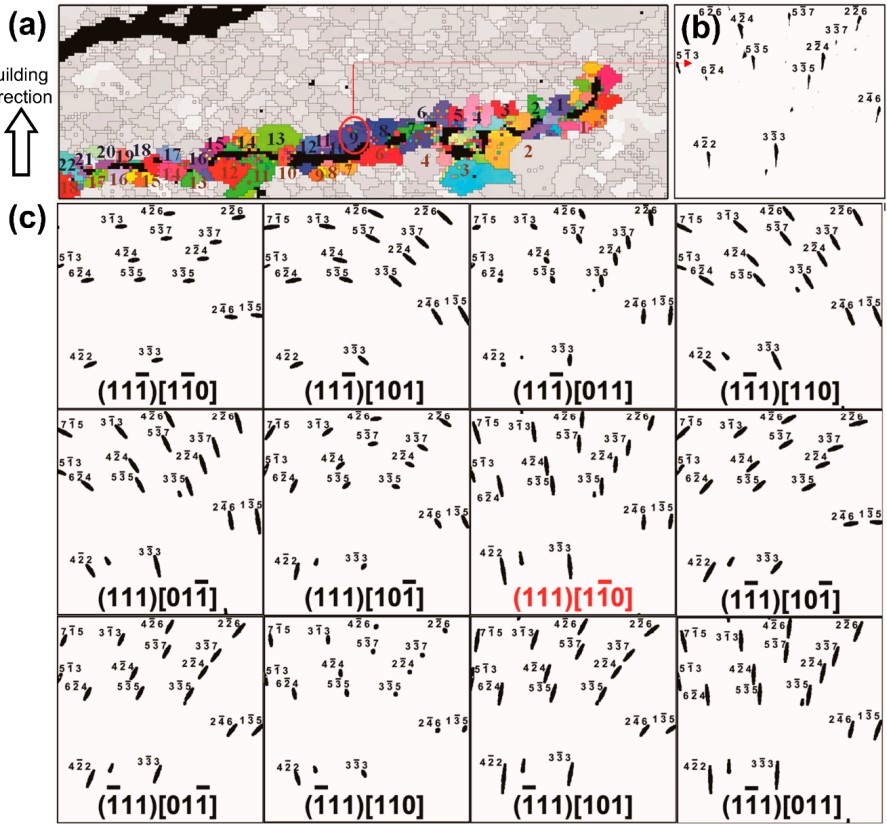

**Figure 8.** (**a**) Numbered grain map around the ductility dip crack selected for the study; (**b**) The experimental Laue diffraction pattern taken from grain number 9; (**c**) Simulations of the Laue peak streaking direction corresponding to all the 12 possible {111}/⟨110⟩ slip systems of the face-centered cubic (FCC) structure. The one highlighted in red is the best match of the simulated shapes with pattern shown in (**b**). The scanned area is 400 μm by 130 μm and the scanning step size is 2 μm (Adapted from [66]).

## 8. Summary and Perspective: Towards Quantitative Scanning Laue Diffraction Microscopy

Laue X-ray microdiffraction is already a powerful tool for the study of the microstructure and mechanical properties of materials at the submicron scale. Like EBSD, Laue X-ray microdiffraction data yields an array of microstructural information which can be directly used to test deformation models. These include crystal orientation, elastic strain tensor, dislocation densities, and identification of dislocation slip systems. However, Laue X-ray microdiffraction offers the distinct advantage of not requiring time-consuming sample preparation and works in a variety of sample environments, while EBSD measurements are confined to an ultra-high vacuum. EBSD has the edge on higher spatial resolution (in the tens of nanometers) and data analysis speed. However, similarly to the trend of X-ray microscopy catching up with electron microscopy with the development of such techniques as synchrotron X-ray ptychography [67], Laue X-ray microdiffraction is closing in on EBSD with respect to performance. New synchrotron beamlines such as 21A at the TPS already provide routine spatial resolution of less than 100 nm, and X-ray beam focus of down to a few tens of nanometers has already been demonstrated with achromatic focusing [68].

The penetrating power of the hard X-ray beam is often seen as a distinct advantage over electron probes for the reasons already mentioned above. However, it has also one crucial drawback: everything which is in the path of the beam below (and above) the sample surface and up to the length of X-ray penetration will either diffract or contribute to the background signal in some ways. This often results in overly complex Laue patterns to analyze in which multiple crystals can diffract simultaneously. A good

percentage of calculation time involved in Laue indexing is in the sorting of reflections belonging to several grains (multigrain analysis). Laue diffraction analysis codes such as XMAS are continuously updated to be able to find analytical solutions to increasingly complex situations while maintaining the need for real-time analysis. Future software developments will increasingly rely on algorithms developed in the field of artificial intelligence (AI). The new indexing algorithm in XMAS, for instance, uses a modified version of a pattern recognition algorithm used for fingerprint matching [69], resulting in enhanced indexing speed for low symmetry materials. The Laue sorting routine in XMAS uses this algorithm to map multiphase materials where Laue pattern indexing is attempted for several crystalline structures provided in a list of possible crystalline phases. For unknown phases or in the absence of a list of possible structures, the future planned automation of the ab initio indexing routine [42] will provide similar answers. The XMAS adaptive indexing routine is used to track reflections of the underlying single crystalline silicon layer beneath a polycrystalline metallization layer to map the metallization induced strain field in thin silicon solar panels [70,71]. The use of machine learning algorithms is also being tested to sort Laue patterns according to crystal orientation and phases.

Technical developments include the quest for depth sensitivity, i.e., 3D Laue microdiffraction. The APS ID-34 beamline has been used to pioneer a technique called differential aperture X-ray microscopy (DAXM) [30] which consists in using a thin wire scanned near the surface of the sample to probe the origin of each diffracted beam in the path of the incident polychromatic beam through the sample. The analysis of the depth-reconstructed Laue patterns provides a 3D map of crystal orientation and strain in the sample. An alternate approach is pencil beam Laue diffraction tomography [72], which consists in using the tomographic approach to 3D reconstruction. Both techniques are rather time-consuming in terms of data collection and currently challenging to be fully implemented for widespread use across several synchrotron sources. However, iterative progress both in terms of data collection strategy (the APS, for instance, is experimenting with accelerated DAXM data collection using multiple wires [73]) and data reconstruction will eventually turn the tide and make these instruments more routinely available.

Another area of future development for the Laue microdiffraction technique is full energy resolution. While the energy of particular reflections can be determined independently by inserting and scanning a monochromator or a rainbow filter in the path of the X-ray polychromatic beam, these measurements add to the data collection time and do not provide a complete picture. Laue patterns obtained by current detector technology have no or very limited energy resolution. However, if the energy of each photon hitting the detector was to be measured, an additional level of information could be extracted from the scan performed to obtain Figure 2 (calculated by averaging the intensity of each Laue pattern taken at each position in the map) for instance. Energy resolution would not only identify the elements contained in each precipitate directly from the background X-ray fluorescence signal but determine the absolute values (as opposed to the relative values provided by Laue patterns alone) of lattice parameters. The dilatational strain field could also be quantitatively measured. The merging of Laue diffraction with spectroscopy has been recently demonstrated with the prototypic pixelated energy-dispersive MAIA array detector [74]. The current MAIA detector consists only of 384 silicon diodes, but increasing miniaturization and computing resources will likely result in a new generation of pixel detectors with the energy-resolving capabilities suitable for Laue diffraction. Energy resolution has also been demonstrated using direct detection fast CCD [75] in photon-counting mode and is able to correctly assign energy to each Laue peak [42].

As brighter X-ray sources and faster, noiseless detectors will contribute to greatly diminish data collection time and powerful computer and new algorithms based on artificial intelligence will provide real-time processing of complex datasets, the Laue X-ray microdiffraction technique is destined to have a bright future as it evolves into a standard quantitative microstructural imaging tool with nanometer resolution.

**Author Contributions:** Conceptualization, K.C. and N.T..; Methodology, N.T.; Software, J.K., W.Z. and N.T.; Validation, G.Z., Y.L., J.K., W.Z., K.C. and N.T.; Formal Analysis, G.Z., Y.L., J.K. and W.Z.; Investigation, G.Z., Y.L., J.K. and W.Z.; Resources, K.C. and N.T.; Data Curation, G.Z., Y.L., J.K., W.Z., K.C. and N.T.; Writing-Original Draft Preparation, N.T.; Writing-Review & Editing, G.Z., K.C. & N.T.; Visualization, G.Z., Y.L., J.K. and W.Z.; Supervision, K.C. and N.T.; Project Administration, K.C. and N.T.; Funding Acquisition, K.C. & N.T.

**Funding:** This research was funded by the US DOE Office of Science User Facility under contract No. DE-AC02-05CH11231, the National Natural Science Foundation of China under Grant No. 51671154, the National Key Research and Development Program of China, under Grant No. 2016YFB0700404, the National Basic Research Program of China ("973" Program) under Grant No. 2015CB057400, the International Joint Laboratory for Micro/Nano Manufacturing and Measurement Technologies, and the Collaborative Innovation Center of High-End Manufacturing Equipment.

**Conflicts of Interest:** The authors declare no conflict of interest.

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
