# Peer review of "Quantitative Scanning Laue Diffraction Microscopy: Application to the Study of 3D Printed Nickel-Based Superalloys"

_qubs, doi:10.3390/qubs2020013_

Round 1

Reviewer 1 Report

This is a nice review of the current state of the art at the ALS microlaue beamlines. There are  many minor points the authors will surely wish to address (below), but no "show-stoppers" were found.

- English language could be polished. 
        first sentence of abstract is too long
        L69 "there is many" -> "there are many"
        L83 symbols garbled for phases on my PDF printout.
        L162 "...sample environment  ... to the next level prefiguring where ..." Not sure you mean prefiguring ?

        etc

- Introduction: laue techniques are dominant in neutron single crystal diffraction, especially for biological systems but also for chemical crystallography, so neutrons deserve a mention also.

- Not sure where, but missing in the manuscript: The actual number of harmonic overlaps (h,k,l) with (h/2,k/2,l/2) etc is rather small. Using a PILATUS detector you should be able to fiddle with the threshold to discriminate these as it advertises 500 eV resolution. Did you do that and does it work? My understanding is that these overlaps are not a problem anyway.

- L183 Frame rate is 20 Hz. Is this photon flux limited or due to the detector saturation? I am aware of 200Hz for various PILATUS experiments. What would increase the collection speed, better detectors or more photons? Any issues with radiation damage?

- Around L206 - the Bragg peak to background ratio is MUCH better for monochromatic beam. This is one of the real reasons why monochromatic is often preferred (going back to earlier part of introduction): you can use smaller crystals that have lower quality with monochromatic beam.

- Figure 2 caption is wrong (repeats figure 1)

- L225: unwanted line break.

- L249: "... inside a diamond anvil cell can be probed without difficulty". Please add a citation. While this is subjective, I disagree with the words "without difficulty". Dealing with a high pressure cell is difficult for most people for a variety of reasons, not limited to:
    - limitation of opening angle
    - parasitic secondary incident beams from diamonds
    - extinction in both incident and diffracted beams
"Possible" and "easy" are not the same thing.

- L295: use of GPU's and cloud computing should also be interesting in the future. Happy case for Laue processing is that each image can be treated independently so should run well in parallel.

- L321: Please cite the "rainbow filter" device at BM32/ESRF for spot-energy determination.

Conclusions: A number of synchrotrons around the world are upgrading or being built to improve the brilliance with MBA rings, but I am under the impression that this does not help much for white beam beamlines as the total flux is about the same for wigglers and bending magnets. Further improvements on detectors and data reduction seem more likely to help. My impression is that micro-laue experiments are rather limited on sample enviroments. For example, the strains in nickel based superalloys are somewhat more interesting under working conditions: so under fatigue loading and at high temperature. Such conditions simply do not fit with the limited space and fragile high resolution mechanics... perhaps my impression is wrong?

I enjoyed the review and recommend publication with the expectation that the authors will resolve the issues mentioned above. 

Author Response

Please see file attached.

Reviewer 2 Report

The paper by Zhou et al. is a review of recent advances of the Laue microdiffraction capabilities at the synchrotron ALS in Berkeley. Data obtained on a Ni-based superalloy elaborated by additive manufacturing is shown as illustrative examples. The paper goes through the different microstructural information that the technique can deliver. It also makes a specific focus on ongoing and future developments concerning software developments to deal with the huge amount of digital information generated.

The paper is well written and clearly organized. The topic matches with the special issue Strain, Stress and Texture Analysis. Therefore, I suggest accepting the paper after minor revision:

1/ The legend of fig 2 is not correct (it is the one of figure 1)

2/ In the pdf version I got, in section 2, the symbol corresponding to \gamma and \gamma’ have been changed into weird @.

3/ In figures 4, 5, 6, it would be nice if the authors could show more clearly where are the substrate and the HAZ.

4/ line 317, it could be indicated that even better resolution (down to 10^-5) can be obtained when Digital Image Correlation technique is used to estimate the distortion of the Laue pattern, see J. Petit, O. Castelnau, M. Bornert, F. Zhang, F. Hofmann, A.M. Korsunsky, D. Faurie, C. Le Bourlot, J.S. Micha, O. Robach, O. Ulrich, Laue-DIC: a new method for improved stress field measurements at the micron scale, J. Synchr. Rad., 22, p.980-994, 2015.

5/ lines 326-330: the building up of a tensile stress perpendicular to the building direction is not straightforward… The authors could provide some more details about the underlying physical processes than just “likely generated by the rapid solidification process”.

Author Response

Please see file attached.

Reviewer 3 Report

My comments to the authors are given in the attached file.

Yours sincerely,

Author Response

Please see file attached.
